# Mechanisms of Acquired Resistance and Tolerance to EGFR Targeted Therapy in Non-Small Cell Lung Cancer

**DOI:** 10.3390/cancers15020504

**Published:** 2023-01-13

**Authors:** Houssein Chhouri, David Alexandre, Luca Grumolato

**Affiliations:** Univ Rouen Normandie, Inserm, NorDiC UMR 1239, 76000 Rouen, France

**Keywords:** non-small cell lung cancer, targeted therapy, EGFR-TKI, drug resistance, drug tolerant persister cells, cellular barcoding

## Abstract

**Simple Summary:**

Lung cancer is the major cause of cancer-related deaths worldwide. The development of targeted therapies has dramatically improved the outcome of lung cancer patients. However, despite an initial response, tumors almost invariably relapse as a result of acquired resistance. In this review, we discuss how lung cancer cells become resistant or tolerant to targeted therapy.

**Abstract:**

Non-small cell lung cancers (NSCLC) harboring activating mutations of the epidermal growth factor receptor (EGFR) are treated with specific tyrosine kinase inhibitors (EGFR-TKIs) of this receptor, resulting in clinically responses that can generally last several months. Unfortunately, EGFR-targeted therapy also favors the emergence of drug tolerant or resistant cells, ultimately resulting in tumor relapse. Recently, cellular barcoding strategies have arisen as a powerful tool to investigate the clonal evolution of these subpopulations in response to anti-cancer drugs. In this review, we provide an overview of the currently available treatment options for NSCLC, focusing on EGFR targeted therapy, and discuss the common mechanisms of resistance to EGFR-TKIs. We also review the characteristics of drug-tolerant persister (DTP) cells and the mechanistic basis of drug tolerance in EGFR-mutant NSCLC. Lastly, we address how cellular barcoding can be applied to investigate the response and the behavior of DTP cells upon EGFR-TKI treatment.

## 1. Introduction

Lung cancer is one of the most commonly diagnosed tumors and the leading cause of cancer-related deaths worldwide, claiming an estimated 1.8 million lives annually [1]. Histologically, lung cancer can be divided into two groups: small-cell lung cancer (SCLC) and non-small cell lung cancer (NSCLC). SCLCs constitute approximately 15% of the cases and are believed to arise from neuroendocrine cells in the airways. They are characterized by the expression of common neuroendocrine markers, such as synaptophysin and chromogranin A. SCLC is a very aggressive malignancy with early and frequent metastases, and it is strongly related to cigarette smoking. Bi-allelic inactivation of tumor suppressors TP53 and RB1 are the most frequent genetic alterations in this type of malignancy [2].

NSCLC represents the remaining 85% of lung cancers and includes three major histological subtypes: adenocarcinoma, squamous cell carcinoma, and large cell carcinoma [3]. Adenocarcinoma is the most common type of NSCLC, accounting for 40-50% of the cases. It usually develops in the periphery of the lungs and originates from type II alveolar cells, which secrete mucus and other substances. These tumors show features of glandular differentiation and are characterized by the expression of the thyroid transcription factor 1 and cytokeratin 7 [4]. Squamous cell carcinoma is responsible for about 30% of NSCLCs. This type of cancer arises most frequently in the proximal bronchi and it generally has the strongest association with smoking [5]. Large cell carcinoma is the least frequent subtype of NSCLC. It may originate anywhere in the lungs and tends to grow quickly. This type of carcinoma is relatively undifferentiated, and it is diagnosed by exclusion. The criteria changed with the new classification published in 2015 by the World Health Organization, and the proportion of large cell carcinomas dropped from about 10% of the cases to the low single digits [6,7,8].

## 2. Targeted Therapy

In the last twenty years, the identification of key genetic events driving tumor growth and survival has dramatically redefined the treatment of NSCLC based on their molecular characteristics [9]. These genetic aberrations occur in certain oncogenes and can serve as drug targets. This is mainly due to the dependency of certain tumors on a single dominant oncogenic protein or pathway to sustain their proliferation [10]. Inhibition of this specific oncogene can be sufficient to induce substantial growth arrest, resulting in tumor shrinkage. Many driver mutations have been identified in NSCLC (Figure 1). The most common are represented by activating mutations of the Kirsten rat sarcoma viral oncogene homolog (KRAS) and the epidermal growth factor receptor (EGFR), observed in 30% and 15% of the patients, respectively. Several other genetic aberration potentials have been identified at lower frequencies in NSCLCs, including translocations involving the anaplastic lymphoma kinase or the ROS1 proto-oncogene receptor tyrosine kinase, including amplification or mutations of the MET tyrosine kinase receptor and point mutations of the BRAF serine/threonine kinase [9]. In recent years, drugs targeting these pathways have been developed, and some have been clinically approved for NSCLC patients. Because of their high response rates and increased specificity compared to standard chemotherapy, targeted agents are used in the first line for the treatment of certain types of patients.

## 3. EGFR Mutations in NSCLC

EGFR belongs to the ErbB family of receptor tyrosine kinases (RTKs) and binds ligands such as EGF, transforming growth factor alpha (TGF-α) and amphiregulin. The ErbB family consists of four related members, EGFR/ErbB1/HER1, ErbB2/HER2, ErbB3/HER3, and ErbB4/HER4, which are implicated in a wide range of biological processes [13]. Under physiological conditions, ligand binding to EGFR via its extracellular domain triggers receptor homo- and/or heterodimerization with other ErbB members, resulting in auto- and transphosphorylation of the intracellular domain on tyrosine residues. The activated receptors signal to downstream pathways, including the canonical mitogen-activated protein kinases (MAPK), PI3K/AKT/mTOR, and JAK/STAT, to regulate cell proliferation and survival. Somatic activating mutations of EGFR have been reported in 15% of NSCLC patients (ranging from 10 to 50% depending on the population, with higher frequencies in patients of East Asian ethnicities [14]), and are responsible for constitutive ligand-independent receptor signaling [11,15,16]. These mutations are generally found in adenocarcinomas, and they provoke a conformational change that shifts the tyrosine kinase domain towards an active state. Exon 19 deletions and L858R substitution represent 90% of EGFR activating mutations in NSCLC [17,18]. Other less common aberrations have been reported, including G719X, L861X, and S768I substitutions and exon 20 insertions [19,20].

## 4. EGFR Targeted Therapy

The presence of these EGFR-activating mutations has been associated with dramatic responses to treatment with EGFR-TKIs in patients with advanced NSCLC [21]. Different types of EGFR-TKIs have been developed, and some of them have been clinically approved, with response rates ranging from 50 to 80% [11] (Table 1).

### 4.1. First Generation EGFR-TKIs

The first generation of EGFR-TKIs is constituted of compounds, such as gefitinib and erlotinib, that reversibly compete with ATP for the binding to the tyrosine kinase pocket of the receptor. These drugs demonstrated improved progression-free survival (PFS), overall survival, and overall response rates (ORR) compared to standard chemotherapy in patients with mutant EGFR NSCLC [22,23,24]. Based on these data, gefitinib and erlotinib were approved as a first-line treatment for NSCLC patients harboring EGFR sensitizing mutations. Unfortunately, not all the patients respond to these inhibitors, and those who respond to the treatment almost invariably relapse, as the tumors become resistant to EGFR-TKIs. Resistance to targeted therapy falls into two main categories: primary and acquired resistance.

Primary resistance relates to tumors that fail to respond to the treatment. For example, it has been shown that NSCLCs containing certain activating mutations of EGFR, such as exon 20 insertions, and they are insensitive to most EGFR-TKIs [25]. Primary resistance can also be due to the presence of other concurrent aberrations, such as in the case of a deletion of BIM (also known as BCL2L11), which has been shown to decrease TKIs response in patients harboring EGFR sensitizing mutations [11,26].

Acquired resistance arises in patients after an initial period of drug response, and it can be classified as on-target or off-target. On-target resistance is caused by secondary EGFR mutations that affect the binding of the inhibitor to the receptor (Figure 2A). The most common mechanism of acquired resistance to first generation, reversible EGFR-TKIs is the emergence of the T790M gatekeeper mutation in the ATP binding pocket of EGFR, occurring in 50–60% of the patients that relapse after an initial response to gefitinib or erlotinib [27,28]. It has been shown that this mutation prevents drug binding by increasing the affinity of the mutant receptor for ATP, thus reducing the potency of ATP competitive EGFR-TKIs [29]. In other patients, off-target resistance results from events that can bypass EGFR signaling, including amplification of other RTKs such as MET, downstream activation of certain components of the pathway or histological/phenotypic transformation [11] (Figure 2B).

### 4.2. Second Generation EGFR-TKIs

Second generation inhibitors, including afatinib and dacomitinib, were originally designed in the hope to overcome the T790M-mediated resistance. Unlike gefitinib and erlotinib, these agents are irreversible inhibitors that also target HER2 and HER4 and have shown promising activity against the EGFR-T790M mutation in preclinical models. However, despite encouraging in vitro data, these EGFR-TKIs failed to prevent the emergence of this mutation in patients due to non-selective inhibition of wild-type EGFR [30,31]. In clinical trials, the efficacy of these drugs was similar to that of first generation EGFR-TKIs, with some differences in toxicity [32,33]. Afatinib and dacomitinib were approved for metastatic NSCLCs harboring non-resistant EGFR mutations, and they can be used as an alternative to first generation inhibitors.

### 4.3. Third Generation EGFR-TKIs

Given the limited efficacy of second-generation EGFR-TKIs in overcoming T790M resistance in NSCLC patients, a number of third generation inhibitors were developed. These agents can form an irreversible covalent bond with the cysteine-797 residue in the ATP binding site of EGFR and showed potent activity against the EGFR-T790M mutation, while poorly inhibiting the wild-type receptor. The first third generation EGFR-TKIs reported was WZ4002, identified by screening of a library of irreversible kinase inhibitor specifically targeting the T790M mutant EGFR. This drug was found 100-fold less potent against wild-type EGFR and 30 to 100 fold more potent against EGFR-T790M [34]. While WZ4002 did not progress into clinical trials, other EGFR-TKIs with similar characteristics were developed and taken into early phase clinical trials. Among these, rociletinib and osimertinib were the first compounds to show significant clinical activity in EGFR mutated NSCLC patients who had relapsed after erlotinib, gefitinib, or afatinib treatment [35,36]. While rociletinib was finally discarded because of side toxicity (it also inhibits the insulin receptor, provoking hyperglycemia) and lower efficacy [37], the FDA approved osimertinib in 2015 for the treatment of patients with metastatic NSCLCs having progressed on first/second generation EGFR-TKIs through the emergence of the T790M mutation. The phase I/II AURA trial evaluated the pharmacokinetic profile, safety, and efficacy of osimertinib in EGFR mutant patients progressing to prior EGFR-TKIs therapy and initially demonstrated promising results with response rates over 70% and median PFS of 9.6 months when only T790M positive patients were considered [38]. The phase III AURA3 study further examined osimertinib compared to platinum/pemetrexed chemotherapy in 419 T790M-positive advanced NSCLC patients who had progressed to first generation EGFR-TKIs. In this trial, osimertinib was associated with higher response rates and median PFS (mPFS) compared to chemotherapy (ORR 71% vs. 31%; mPFS 10.1 vs. 4.4 months), thus establishing osimertinib as the standard of care in this setting. Median PFS benefit was also seen in patients with brain metastases (8.5 vs. 4.2 months) [39]. In addition, the phase III FLAURA study was designed to compare osimertinib to first generation EGFR-TKIs in front-line therapy for EGFR-mutated, treatment-naïve NSCLC patients. In this trial, osimertinib demonstrated prolonged PFS (18.9 months vs. 10.2 months) and median overall survival (38.6 months vs. 31.8 months) compared to gefitinib or erlotinib [40,41]. Based on these results, osimertinib was approved as first-line therapy for NSCLC patients harboring EGFR activating mutations. Other third generation EGFR-TKIs, such as almonertinib and lazertinib have been approved in China and South Korea, respectively. These inhibitors have demonstrated good efficacy and safety in patients with EGFR-T790M positive NSCLC [42,43].

Despite its efficacy, resistance to osimertinib inevitably develops. Similar to earlier generations, resistance to third generation EGFR-TKIs is classified as primary and acquired. EGFR-C797S mutation represent the most commonly occurring on-target resistance mechanism on osimertinib treatment [44,45]. This mutation replaces the residue covalently bound by osimertinib, thus dramatically reducing the efficiency of kinase inhibition. The C797S mutation is seen in approximately 10–20% of NSCLC patients at disease progression to second-line osimertinib [44,46], and it can emerge in cis or in trans with the EGFR-T790M mutation. The coexistence of C797S and T790M on the same allele (in cis) confers resistance to all generations of EGFR-TKIs, while when these mutations are on different alleles (in trans), the tumors retain sensitivity to the combination of first and third generation EGFR inhibitors [47]. In the front-line setting, C797S occur in 7% of patients and has been shown to emerge in the absence of T790M, in which case the tumor remain sensitive to first generation EGFR-TKIs [48]. In addition to C797S, other, less frequent on-target EGFR mutations that interfere with the drug binding have also been reported as mechanism of resistance to osimertinib, including G796R/S/, L792H, L718Q, and G724S substitutions [49,50]. It is worth noting that loss of T790M has been observed after relapse to osimertinib treatment. This suggests that T790M-positive clones could co-exist with other resistance mechanisms, which is associated with poor responses and shorter PFS [51,52]. As described above for first-generation EGFR-TKIs, off-target mechanisms of resistance to osimertinib are frequently observed in patients, with MET amplification being the most common (15%) [52]. Histological transformation to SCLC has also been reported in some cases [53,54].

### 4.4. Fourth Generation EGFR-TKIs

In order to overcome C797S-mediated acquired resistance, fourth generation EGFR-TKIs were developed. EAI045 is an allosteric, non-ATP competitive inhibitor targeting both the T790M and C797S mutations. It binds to the allosteric sites on EGFR, created by the displacement of the regulatory C-helix in the inactive conformation of the kinase. Due to EGFR dimerization, EAI045 is not effective as a single agent, and it requires the co-administration of the anti-EGFR antibody, cetuximab [55]. Recently, JBJ-09-063 was reported as a new EGFR allosteric inhibitor that is effective as a single agent in models harboring EGFR L858R/T790M/C797S mutations [56]. Beside allosteric inhibitors, several fourth generation EFR-TKIs that can covalently bind to EGFR have been reported, such as UPR1444, which potently and irreversibly inhibits the EGFR-L858R/T790M/C797S through the formation of a sulfonamide bond with the catalytic residue Lys745 [57]. It is also worth noting that several clinical trials are currently ongoing to evaluate the clinical efficacy and the safety profiles of fourth generation EGFR inhibitors, including BLU945 [58], BBT176 [59], and TQB3804 [60].

### 4.5. EGFR Degraders

Fourth generation EGFR-TKIs appear to be potent and effective against the EGFR-C797S mutations and to have strong antitumor activity in preclinical models. Besides TKIs, other types of inhibitors have been designed that could improve the treatment of patients with EGFR-mutant NSCLC. In particular, the discovery of EGFR degraders holds great promise. A key focus of targeted protein degradation is the development of proteolysis targeting chimeras (PROTACs). PROTACs are heterobifunctional small-molecule degraders, typically consisting of two linked moieties, with one binding the protein of interest and the other binding an E3 ligase. PROTACs recruit the E3 ligase to targeted protein, leading to its selective ubiquitination and degradation by the proteasome [61,62]. Multiple selective EGFR degraders were developed, which can selectively inhibit the proliferation of EGFR-mutant NSCLC cells [63,64,65]. Recently, Du et al. described a novel EGFR-based PROTAC, HJM-561, that potently inhibits the proliferation of tumor cells harboring the EGFR-C797S mutation [66].

## 5. Drug Tolerant Persister States

Despite the high response rates of EGFR-TKIs, acquired resistance almost invariably occurs. In some cases, this is due to the selection of pre-existing cells harboring well defined genetic resistance mechanisms, such as the T790M mutation or MET amplification, that enable growth during the treatment [67,68,69]. As an alternative mechanism, it has been shown that resistance can also arise from drug-tolerant persister (DTP) cells, sometimes referred to as minimal residual disease, which can survive during treatment, when the large majority of the cancer cells dies. In the presence of the drug, these subpopulations display a slow proliferative rate with altered cellular metabolism, and can survive prolonged treatment through epigenetic adaptations [70,71].

### 5.1. Persisters in NSCLC

The term of DTP comes from the field of microbiology, where it has been shown that a small fraction of dormant bacteria has the capacity to survive in the presence of antibiotics despite the fact that they do not contain a genetic mechanism of resistance. These drug tolerant cells are able to resume growth and re-establish a drug sensitive population upon drug withdrawal, indicative of a transient and not inheritable resistance mechanism [72]. Similar to bacterial persisters, cancer cells can enter a reversible drug-tolerant state when exposed to anticancer therapy [73]. The first report describing the transient acquisition of drug-tolerant state in NSCLC came in 2010 from Jeffrey Settleman’s laboratory. In this study, Sharma and colleagues used the well established EGFR-mutant NSCLC cell line PC9 and observed that a small fraction (range 0.3–5%) of quiescent cells remained viable after nine days of treatment with the first generation EGFR-TKI erlotinib. These DTPs can resume normal proliferation in the presence of gefitinib or erlotinib and give rise to a second population of cells termed drug-tolerant expanded persisters (DTEP). Of note, the authors showed that, after drug withdrawal, these populations remained resistant to EGFR-TKIs for up to 90 cycles of cell divisions, before reverting to a drug-sensitive state [74]. After this initial description, other studies showed that DTPs can function as a reservoir from which heterogeneous mechanisms of acquired resistance can arise [68,75]. Similar subpopulations of drug-tolerant cells have also been identified in other types of cancer, including colon cancer [76], melanoma [77], and glioblastoma [78].

### 5.2. Molecular Characteristics of Drug Tolerance in NSCLC

Several mechanisms have been associated with the ability of drug-tolerant cells to withstand EGFR-TKIs treatment, including chromatin remodeling, activation of bypass pathways, and altered cellular metabolism. In this section, we will summarize the main characteristics of drug-tolerant state described in EGFR-mutant NSCLC. For a more general discussion of how DTP cells can escape cell death in different types of cancer, please refer to our recent review [79].

#### 5.2.1. Epigenetic Modifications

To gain insight into the underlying molecular mechanism of the drug-tolerant phenotype, Sharma and colleagues analyzed the gene expression profiles of DTP and DTEP and found that these populations are characterized by altered chromatin states. In particular, the authors showed that these cells were able to survive in the presence of EGFR-TKIs by up-regulating the histone demethylase KDM5A, while they could be selectively ablated by the histone deacetylase (HDAC) inhibitor trichostatin A. The study also suggested a role for the insulin-like growth factor 1 receptor (IGF-1R) in the emergence of DTP populations: activation of this receptor was shown to drive drug tolerance by increasing the expression of KDM5A, leading to overall repressive changes in chromatin structure [74].

In line with these findings, Guler et al. showed that EGFR-TKIs induce increased expression of the long-interspersed repeat element 1 (LINE-1), an active retrotransposable element that can propagate and insert randomly throughout the genome, resulting in genome instability. The authors showed that DTPs exhibit a repressive chromatin state by increased methylation of lysines 9 and 27 of histone H3 (H3K9 and H3K27), particularly on the LINE-1 locus, thus decreasing DNA damage in this cell sub-population. Treatment with histone methyltransferase inhibitors, such as tazemetostat, which specifically reduces the global levels of H3K27me3 by inhibiting the activity of enhancer of zeste homolog 2, increases the chromatin accessibility and results in the ablation of DTP cells through derepression of LINE-1 elements [80].

#### 5.2.2. Reactivation of EGFR Signaling and Up-Regulation of Other Pathways

Reactivation of ERK1/2 has been identified as a resistance mechanism to EGFR-TKIs in NSCLC. Ercan and colleagues found that this activation is caused by either an amplification in chromosome 22 harboring the mitogen-activated protein kinase 1 (MAPK1) or by down-regulation of negative regulators of ERK signaling, including the dual specificity phosphatase 6 (DUSP6) [81]. A combination of EGFR and MEK inhibitors effectively prevents reactivation of ERK1/2 and delay the emergence of DTP cells [82].

The tyrosine kinase receptor AXL has been reported to mediate NSCLC tolerance in response to osimertinib treatment. Taniguchi et al. showed that AXL can be activated through suppression of a negative feedback loop involving SPRY4, resulting in AXL heterodimerization with EGFR or HER3. Consistent with these findings, inhibition of AXL could restore the sensitivity to osimertinib and prevent the emergence of DTP cells [83]. Of note, AXL can drive EMT transition in EGFR-mutant NSCLC models with acquired resistance to erlotinib [84]. According to a recent study, integrin β3 may stimulate the expression of AXL in EGFR-mutant NSCLC cells [85]. In the presence of EGFR-TKIs, AXL and its ligand GAS6 were shown to promote mutagenesis in DTP cells by inducing the expression of error-prone DNA polymerases and enhancing purine metabolism, thus favoring the emergence of de novo resistance mutations, such as EGFR-T790M [86].

Shah and colleagues found that aurora kinase A (AURKA) plays a major role in the emergence of DTP cells induced by EGFR-TKIs treatment. AURKA activation, which is triggered by its regulator TPX2, prevents osimertinib-induced apoptosis by increasing BIM phosphorylation. The authors showed that targeting AURKA and EGFR could reduce the proportion of DTP in vitro and in vivo [87]. Phase I/II clinical trials are currently ongoing to evaluate the efficacy of this combination in patients with advanced EGFR-mutant NSCLC (NCT04085315/NCT05017025). In addition to AURKA, Tanaka et al. reported a role for AURKB in the emergence of DTP cells in NSCLC. Mechanistically, the authors found that AURKB inhibition overcomes resistance to osimertinib by enhancing BIM and PUMA-mediated apoptosis [88].

In another recent study, it was shown that NSCLC cells survive EGFR and MEK dual inhibition by entering a senescence-like state, which is accompanied by up-regulation of the YAP/TEAD pathway. YAP is a transcriptional coactivator that shuttles between the cytoplasm and the nucleus, where it interacts with the transcriptional factor TEAD and regulates the expression of genes that promote cell growth and survival. The authors found that YAP and TEAD cooperate with the EMT transcription factor SLUG to repress the expression of the pro-apoptotic factor BMF, allowing the cells to escape apoptosis and survive. They also developed a new TEAD inhibitor less toxic compared to previous compounds and they showed that it can enhance apoptosis and prevent the emergence of DTP cells in response to EGFR and MEK dual inhibition [89].

The Wnt/β-catenin signaling has also been associated with the maintenance of DTP cells in response to EGFR-TKIs treatment. One study showed that EGFR inhibition results in the activation of β-catenin signaling in a Notch3-dependent manner, leading to the survival of a subpopulation of DTP cells with stem cell-like properties [90]. Consistent with these data, Maynard and colleagues used single-cell RNA sequencing (scRNA-Seq) to analyze tumor biopsies of EGFR-mutant NSCLC patients treated with osimertinib and showed that DTP cells are characterized by an alveolar-regenerative signature, which was related to activation of Wnt/β-catenin signaling [91].

#### 5.2.3. Metabolic Reprogramming

Remodeling cellular metabolism, including the ability to maintain the redox balance under nutrient-deprived conditions and other stresses, is one of the hallmarks of cancer [92]. Compared to normal cells, which rely primarily on mitochondrial oxidative phosphorylation (OXPHOS) to generate ATP for energy, cancer cells generally depend on aerobic glycolysis. This phenomenon, also known as the Warburg effect, represents the most common feature of metabolic reprogramming observed in cancer cells, and it is characterized by increased glucose uptake via glycolysis, rather than mitochondrial oxidative phosphorylation, regardless of oxygen availability and mitochondrial activity [93]. While aerobic glycolysis is used by rapidly proliferating cancer cells, it has been shown that slowly-cycling DTP cells depend more on mitochondrial respiration for their energy production [71]. This metabolic shift to OXPHOS in DTP cells results in increased levels of reactive oxygen species (ROS). Thus, DTP cells require a robust antioxidant process to protect themselves from oxidative stress [94]. Raha et al. showed that aldehyde dehydrogenase (ALDH) is required for the maintenance of a persistent cell population in NSCLC. They found that ALDH protects DTP cells from ROS-mediated toxicity and that pharmacologic inhibition of ALDH activity leads to accumulation of ROS to toxic levels, causing DNA damage and cell death within the drug-tolerant subpopulation [95]. Consistent with an increased susceptibility to oxidative stress in DTP cells, another study reported that these cells rely on the expression of the glutathione peroxidase 4 (GPX4) in EGFR-mutant NSCLC, as well as in other different types of cancer. The authors found that two GPX4 inhibitors (RSL3 and ML210) were selectively lethal to DTP cells and that GPX4 inhibitor-mediated cell death was accompanied by accumulation of lipid hydroperoxides and could be rescued by the lipophilic antioxidants ferrostatin-1 and liproxstatin-1, suggesting a ferroptotic mechanism [96].

### 5.3. Origin of DTP Cells: Darwinian Selection or Lamarckian Induction?

While it is becoming more and more evident that acquisition of drug resistance does not rely only on genetic mechanisms, a major question that is still unanswered concerns how tolerant/persister cells originate. Certain cell populations could be enriched because of intrinsic epigenetic properties that favor growth in the presence of the drug through a non-genetic Darwinian selection (Figure 3A). For example, Shaffer et al. found that small populations of BRAF-mutated melanoma are primed to become tolerant by transiently expressing high levels of EGFR, AXL, or the nerve growth factor receptor. They showed that these cells exhibit profound transcriptional heterogeneity at the single cell level, which predicts which cells will eventually resist drug therapy [97]. In another study from the same group, the authors performed a high-throughput CRISPR/Cas9 genetic screen to identify modulators of cell fate in the context of resistance to BRAF inhibition in melanoma cells and they found that inactivation of different factors, including DOT1L, LATS2, and BRD2, can modify the proportion of cells primed to become DTP [98]. In a recent preprint from the Raj laboratory, a strategy combining DNA barcoding and scRNA-Seq was used to show that melanoma DTP cells can adopt different transcriptional and functional profiles in response to targeted therapy. By comparing the transcriptional profile of individual drug selected cells with their barcodes across twin replicates derived from the same population, the authors concluded that, for some subpopulations, the DTP phenotype is intrinsically predetermined before the onset of the treatment, suggesting that DTP cells are pre-existing and selected upon drug exposure [99].

According to a different model, the tolerant/persister state could be directly induced by the treatment (Figure 3B). This process, defined as Lamarckian induction [100,101], implies that these cells arise more randomly, through a cell fate decision that can be potentially influenced by stochastic fluctuations of gene expression [102,103] or a particular phase of the cell cycle, as it has been shown for embryonic stem cells [104]. Consistent with this type of scenario, Kurppa et al. reported that NSCLC DTP cells surviving EGFR-MEK dual inhibition do not derive from pre-existing primed clones, but they arise randomly from the mass population of untreated cells [89]. Moreover, gene signatures found in NSCLC DTP cells treated with osimertinib were not present in untreated cells, implying that the induction of the drug-tolerant phenotype is, at least in part, an adaptive process [105]. Rambow et al. identified distinct drug tolerant transcriptional states that emerge upon BRAF/MEK inhibition in melanoma patient derived xenografts. The authors showed that DTP cancer cells displaying a neural crest stem cell-like profile are more likely to give rise to fully resistant clones, indicating that the type of transcriptional changes induced by the treatment can affect the long term fate of DTP cells [106]. Recent studies on colorectal and breast cancer cells suggest that any cancer cells has the ability to enter a drug tolerant persister state in response to chemotherapy by adopting a state that resembles diapause, a highly conserved developmental mechanism used by embryos to survive unfavorable environmental conditions [107,108].

It is important to note that Darwinian selection and Lamarckian induction are not mutually exclusive, since certain subpopulations can be selected based on a particular pattern of gene expression, followed by additional changes induced by the drug. For example, the study by Schaffer et al. discussed above identified rare cell subpopulations displaying particular transcriptional profiles that are primed to survive in the presence of the drug. The treatment induces the acquisition of a more stable resistant phenotype through an epigenetic reprogramming, possibly engendered by inhibition of SOX10-mediated differentiation and induction of AP1 and TEAD transcription factors [97]. A better understanding of the mechanisms underlying the survival of certain cell populations during treatment could provide new strategies to target residual cancer cells, a necessary step to improve clinical efficacy and prevent tumor relapse.

## 6. Cellular Barcoding

Cellular barcoding is a powerful strategy to investigate the mechanisms of resistance to anti-cancer therapies. The basic principle involves the tagging of individual cells of interest with unique and heritable labels. Different strategies have been developed to integrate the barcodes into the genome of the cells using lentiviral vectors or CRISPR/Cas9 technology.

### 6.1. Lentiviral Barcodes to Study Resistance to Anti-Cancer Drugs

To study the fate of cancer cells in response to treatment, lentiviral vectors can be used to efficiently deliver thousands to millions barcodes that are integrated into the genome of the cells. The barcode corresponds to a highly complex stretch of nucleotides that is inserted in a particular region of the vector. The vectors, each containing a different barcode, are pooled to form a library that can be used to transduce the cells of interest. In the case of completely degenerated sequences, the possible combinations are equivalent to 4^N^, where N stands for the number of nucleotides. For example, a 20 bp sequence can yield 4^20^ (~10^12^) unique barcodes. To avoid the generation of aberrant sequences, such as long repeats or highly unbalanced proportions of G/C or A/T, semi-random pools of DNA barcodes can also be designed, in which certain positions are constrained to one or more specific nucleotides. To ensure that each cell contains only one copy of the vector, and hence one barcode, the lentiviral library is transduced at low multiplicity of infection (MOI), followed by selection of the infected cells, generally using an antibiotic, for which the vector encodes a resistance gene. The barcodes can be “read” by amplifying by PCR the corresponding sequence of the vector from the gDNA of the cells, followed by high-throughput sequencing of the amplicon (Figure 4) [109,110].

The viral barcoding strategy has been extensively used over the last few years to perform lineage tracing and to investigate drug resistance in a wide variety of cancers. For example, Bhang et al. developed a high complexity barcode library, named ClonTracer, to individually label several thousand clones within a mass population of NSCLC cells. To assess whether acquired-resistance to EGFR-TKIs is driven by the emergence of pre-existing or de novo clones, they analyzed the barcode composition of different replicates of cells treated in the presence or the absence of erlotinib. They reasoned that, if pre-existing resistant cells are selected during the treatment, a large fraction of shared enriched barcodes should be identified in various replicates. On the other hand, if resistant cells arise de novo during the treatment, distinct barcodes are expected to emerge across replicates. The authors found that 40% of the barcodes were shared in multiple culture replicates, implying that EGFR-TKIs resistant clones can be present before the onset of the treatment [67]. A similar strategy using a different NSCLC cell line was used by the Engelman laboratory to demonstrate that resistant cells can either derive from rare pre-existing clones or from DTP populations that are capable of surviving during the treatment. These cells can then function as a reservoir for the acquisition of de novo mutations that make them fully resistant [68].

### 6.2. Viral-Barcoding Compatible with Single Cell RNA Sequencing

Recent studies described lentiviral barcode libraries that are compatible with scRNA-Seq analysis. To enable the simultaneous detection of the barcode and the transcriptome of individual cells, the lentiviral vectors are designed to contain the highly variable DNA sequence in the 3’UTR of a constitutively expressed transgene. The barcodes are thus expressed and they can be identified by 3’ scRNA-Seq. In the first examples of this type of strategy, the CellTag and LARRY (lineage and RNA recovery) systems, the DNA barcode is located in the 3’ UTR of the GFP mRNA. These libraries were used to investigate the fate of reprogrammed fibroblasts [111] and the different hematopoietic lineages [112]. This strategy has also been employed to investigate drug resistance in cancer cells. Oren and colleagues took advantage of scRNA-Seq and lineage tracing with DNA barcodes to characterize NSCLC DTP cells at single-cell resolution. After two weeks of osimertinib treatment, they identified two main populations of surviving cells: non-cycling DTPs and DTPs that have re-entered cell cycle to divide and form colonies despite drug pressure. To characterize the molecular mechanisms associated with cycling and non-cycling DTPs, the authors developed a system called Watermelon, allowing back-tracing and transcriptional profiling of each cell in the population before and after drug addiction. They showed that these two populations arise from different cell lineages with distinct transcriptional and metabolic programs. Of note, non-cycling DTPs were characterized by the expression of genes associated with cholesterol homeostasis, interferon-α, and notch-signaling. By contrast, the cycling persistent state was characterized by increased expression of the transcription factor NRF2 and decreased levels of ROS. The addition of the ROS scavenger N-acetylcysteine (NAC) was sufficient to increase the proportion of cycling DTPs, consistent with a role of the redox balance in regulating the proliferative ability of these cells. The authors also showed that a switch towards a fatty acid oxidation (FAO) can contribute to the cycling persister phenotype, and that inhibition of FAO using the compound etomoxir reduced the proliferative capacity of DTPs in the presence of osimertinib [105].

Chang and colleagues recently developed a similar approach combining DNA barcoding and scRNA-Seq, called TraCe-seq, to compare the effects of conventional EGFR-TKIs with those of GNE-641, a dual EGFR inhibitor-degrader. They found that GNE-641 was less effective than erlotinib and osimertinib in inhibiting NSCLC cell growth, as well as in reducing the absolute number and the diversity of TraCe-seq barcodes, despite similar levels of MAPK pathway suppression. scRNA-Seq analysis revealed that GNE-641 resistant clones exhibited reduced expression of genes involved in protein processing in the endoplasmic reticulum (ER). The authors showed that the EGFR protein itself plays a crucial role in mediating full cellular efficacy of EGFR-TKIs, as its expression increases ER stress and subsequent pro-death signaling. Consistent with these findings, combination of GNE-641 with low concentrations of ER stress inducers, such as tunicamycin or thapsigargin, strongly enhanced the cytotoxic effects and led to the complete elimination of residual cells. This study uncovered an essential role of the ER protein processing pathway in the response to EGFR targeted therapies [113].

### 6.3. CRISPR-Barcoding

As an alternative strategy of randomly integrating lentiviral libraries, our laboratory developed the CRISPR-barcoding strategy, in which a DNA barcode can be introduced at a specific genome location through CRISPR/Cas9-induced homology directed repair. (HDR). In HDR, a donor DNA co-transfected into the cells functions as a template for precise repair: through appropriate design of the donor DNA, this mechanism can be used to generate a wide range of genetic modifications, including specific point mutations or the insertion of an entire gene. Through CRISPR-barcoding, we inserted a short stretch of degenerated nucleotides in a safe harbor genomic locus of a NSCLC cell line, and we investigated the effects on the clonal architecture induced by EGFR-TKI treatment [114].

## 7. Conclusions

The emergence of resistance is a fundamental cancer property, which mostly derives from the fact that individual tumors are composed of an intricate pattern of heterogeneous subclonal populations, functioning as a complex reservoir that fuels the capacity of tumor cells to adapt to environmental conditions. While most EGFR-mutant NSCLCs initially respond to EGFR-TKIs, they ultimately become resistant due to the emergence of small subpopulations of resistant or tolerant cells. Complex barcode libraries have made it possible to trace the behavior of the individual clones that constitute the mass population of cancer cells, thus providing the means to investigate resistance to anti-cancer therapies under a completely different perspective. A better understanding of the underlying causes of drug tolerance and the acquisition of resistance should help improve the efficacy of cancer treatment.

## Figures and Tables

**Figure 1 cancers-15-00504-f001:**
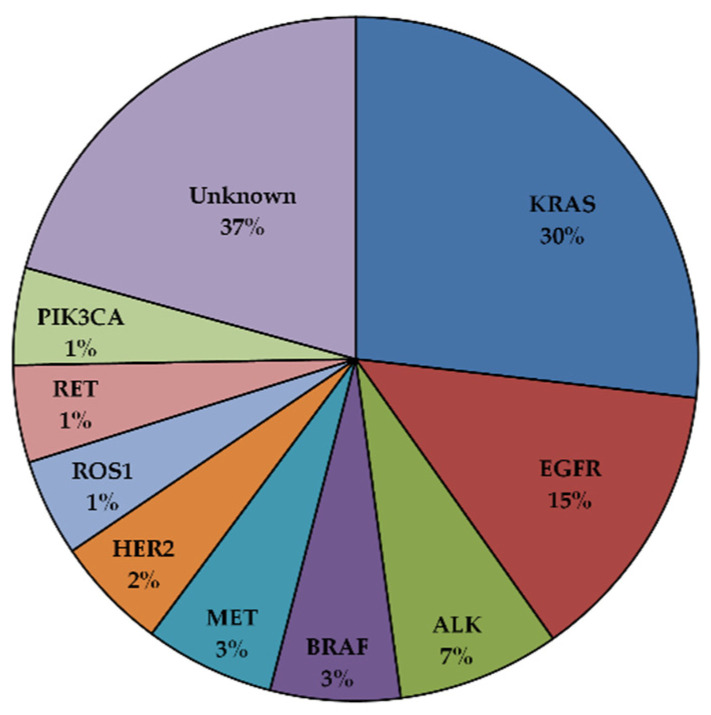
Driver oncogenic mutations in NSCLC [11,12].

**Figure 2 cancers-15-00504-f002:**
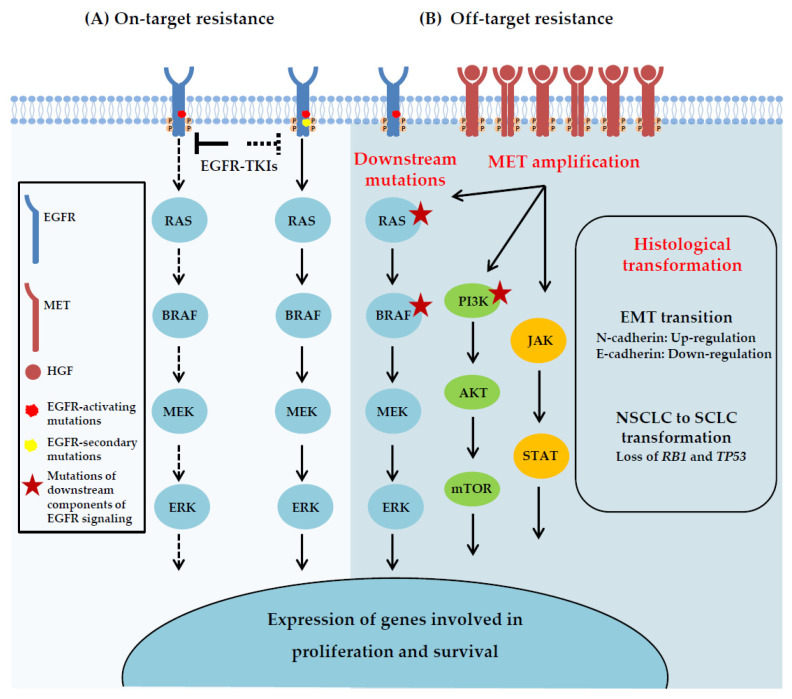
Acquired resistance mechanisms to EGFR-TKIs. (**A**) On-target resistance is caused by second-site EGFR kinase domain mutations that interfere with the binding of the EGFR-TKIs. (**B**) Off-target resistance may result from alterations involving downstream components of the EGFR pathway, activation of alternative signaling pathways that bypass the primary drug targets, including MET amplification, or transition to another cell lineage, such as epithelial-to mesenchymal transition (EMT) and SCLC transformation.

**Figure 3 cancers-15-00504-f003:**
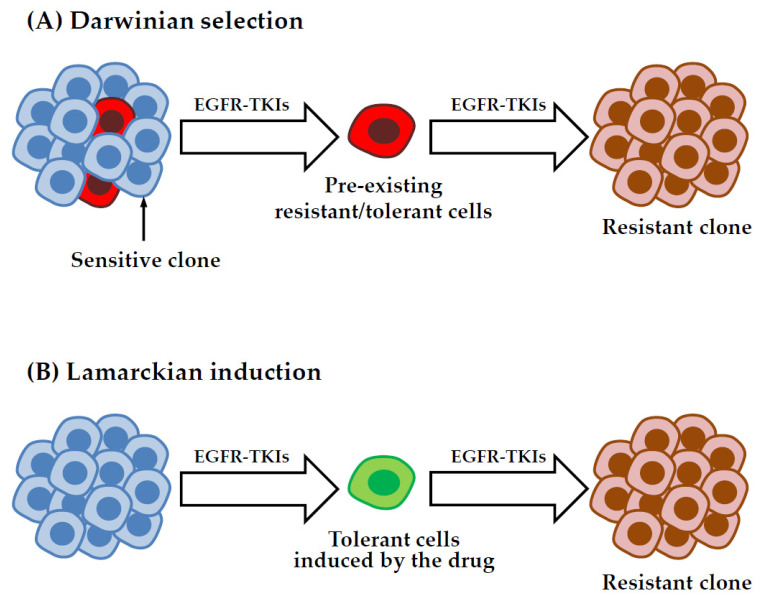
Models for the emergence of drug-tolerant cells. In the pre-existing selection model (**A**), DTP cells are selected in response to treatment because of some intrinsic properties through non-genetic Darwinian selection. Alternatively, in the drug induced model, also known as Lamarckian induction (**B**), DTP cells can originate more randomly as a direct effect of the treatment. These cells can evolve over time to acquire various genetic or non-genetic mechanisms of resistance.

**Figure 4 cancers-15-00504-f004:**
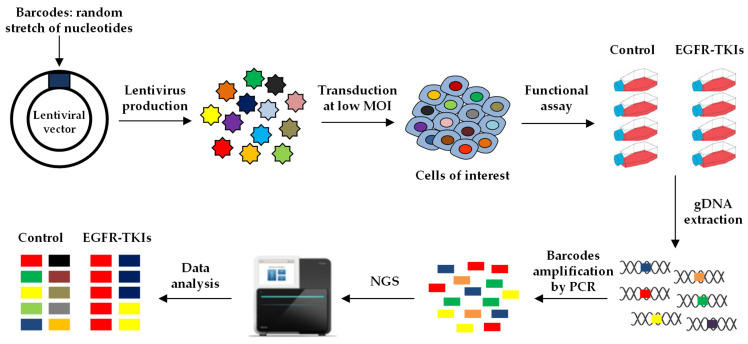
Lentiviral barcoding. A lentiviral barcode library is generated containing a random and highly complex stretch of nucleotides. The virus is produced and used to transduce the cancer cells of interest at low multiplicity of infection to label each cell with one unique barcode. The barcoded cells are treated in the presence or the absence of the compounds of interest, e.g., an EGFR-TKI, and then harvested to extract their genomic DNA (gDNA). The region of the vector containing the barcodes is amplified by PCR and the amplicons are sequenced to assess the relative proportion of each barcode in each sample.

**Table 1 cancers-15-00504-t001:** Different Generations of EGFR-TKIs for NSCLC.

Class	Drugs	Clinical Trial	Approval	CSF Concentration	EGFR Sensitizing Mutations	EGFR Binding
First generation	Gefitinib	NCT01203917	FDA/EMA approved	Low	Del19/L858R	Reversible Competitive
	Erlotinib	NCT00446225	FDA/EMA approved			
	Icotinib	NCT01040780	Approved in China			
Second generation	Afatinib	NCT01466660	FDA/EMA approved	Low	Del19/L858R/ T790M	Irreversible Covalent
	Dacomitinib	NCT01774721	FDA/EMA approved			
Third generation	WZ4002	NA	Preclinical	NA	Del19/L858R/ T790M	Irreversible Covalent
	Rociletinib	NCT01526928	Rejected	Low		
	Osimertinib	NCT02296125	FDA/EMA approved	High		
	Lazertinib	NCT03046992	Approved in South Korea	High		
	Olmutinib	NCT01588145	Approved in South Korea *	NA		
	Avitinib	NCT02330367	Phase I/II (Active)	Low		
	Nazartinib	NCT02108964	Phase I/II (Active)	NA		
	Mavelertinib	NCT02349633	Phase I/II (Terminated)	NA		
	Naquotinib	NCT02588261	Phase III (Terminated)	NA		
	Almonertinib	NCT02981108	Approved in China	Effective in patient with BM	Del19/L858R/ G719X/L861Q/ T790M	
	Alflutinib	NCT03127449	Approved in China			
Fourth generation	EAI001	NA	Preclinical	NA	L858R/T790M/ C797S	Reversible Allosteric
	EAI045	NA	Preclinical	NA		
	JBJ-09-063	NA	Preclinical	NA		
	BLU945	NCT04862780	Phase I/II (Recruiting)	NA	Del19/L858R/ T790M/C797S	Unknown
	BBT176	NCT04820023	Phase I/II (Recruiting)	NA		
	TQB3804	NCT04128085	Phase I (Unknown)	NA		

* Stopped in 2016 because of two cases of toxic epidermal necrolysis with one of them being fatal. EMA: European Medicines Agency, FDA: Food and Drug Administration of the United States, CSF: cerebrospinal fluid; CNS: central nervous system; BM: brain metastases, Del19: exon 19 deletion; NA: not available.

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
