# Peer review of "Mechanisms of Acquired Resistance and Tolerance to EGFR Targeted Therapy in Non-Small Cell Lung Cancer"

_cancers, 2023, doi:10.3390/cancers15020504_

Round 1
Reviewer 1 Report
I went through the manuscript entitled “Mechanisms of Acquired Resistance and Tolerance to EGFR Targeted Therapy in Non-Small Cell Lung Cancer” by Dr Grumolato and co-workers.
Overall, the manuscript is well written and quite pleasant to read, especially the cellular barcoding paragraph, which gives this manuscript a singularity from the many existing reviews on the topic of drug-tolerance to targeted therapy in NSCLC. The references are numerous and relevant, and provide adequate coverage of the area of interest.
Below are the suggestions authors should take into consideration:
1) Line 150-151: Cite the two pivotal studies by Paez, J.G. (Science 2004; PMID: 15118125) and Lynch T.J. (N. Engl. J. Med. 2004; PMID: 15118073), which first reported that the presence of somatic mutations in the kinase domain of EGFR correlated with increased responsiveness to EGFR-TKI.
2) Line 163: reference n°34 refers to the final OS report from the OPTIMAL trial (erlotinib vs chemotherapy), and does not reflect the improved PFS and ORR as stated by the authors. For PFS and ORR benefit of erlotinib vs chemo, the original EURTAC trial would be more appropriated (Rosell et al, The Lancet Oncol 2012; PMID: 22285168).
3) Line 165: “those who”
4) It is unclear how/why Figure 1 is divided into A and B… The authors should probably use color coding to discriminate between on- and off-target resistance in a single figure 1, or alternatively show a clearer separation between both mechanisms.
5) Line 260-261: The sentence “It is worth noting that loss of T790M has been observed after osimertinib treatment” is somehow confusing… By "after treatment with osimertinib" did the authors mean "after relapse to osimertinib treatment"? Loss of T790M is not surprising in response to osimertinib treatment, the interesting point is that T790M may be lost after relapse!
6) Line 395-400: The authors refer to a preprint, therefore non-peer-reviewed article, from Bivona’s lab. Although the scientific rigor and quality of this team is not questioned, it seems premature to cite these data before they have been published in a peer-reviewed journal. This reference should be removed.
7) Line 450-457: Same as before: a preprint article, although it comes from Arjun Raj's lab, should not be cited before it has been published in a peer-reviewed journal. This reference should also be removed.
8) Figure 2 would need several adjustments to improve accuracy: A) arrows on the right are labeled “Resistance” whereas arrows on the left are labeled “EGFR-TKIs”, which does not make much sense… It would appear more meaningful to keep EGFR-TKIs on both sides. B) The authors state that Darwinian selection selects pre-existing tolerant cells, but seem to exclude the possibility of the selection of already resistant cells. I therefore recommend that authors replace "tolerant" with "resistant/tolerant", since current knowledge cannot exclude any hypothesis. C) For both Darwinian selection and Lamarckian induction, the authors indicate that relapsed tumors are “genetically resistant clones”. However, a significant proportion of resistance mechanisms to EGFR-TKIs is non-genetic (e.g. Epithelial-to-Mesenchymal Transition, lineage switch to Squamous, LCNEC or SCLC) and half of the resistant tumors have unknown resistance mechanism, which could therefore be either genetic or non-genetic. The authors should indicate that resistant tumors resulting from Darwinian and Lamarkian selection may also have a non-genetic resistance mechanism, or simply remove the word “genetically”.
9) The cellular barcoding section is probably the most interesting and original part of this manuscript compared to the many existing reviews covering the same topic, and would therefore have deserved a dedicated illustrative figure. While equivalents to figures 1 and 2 have been seen many times, a figure recapitulating the use of cellular barcoding in the context of drug-tolerance/adaptive resistance would bring a special and valuable touch of distinction to this work that would be greatly appreciated.
Author Response
We appreciate Reviewer 1’s positive comments, and we have revised the manuscript in accordance with his/her specific suggestions as follows:
1) Line 150-151: Cite the two pivotal studies by Paez, J.G. (Science 2004; PMID: 15118125) and Lynch T.J. (N. Engl. J. Med. 2004; PMID: 15118073), which first reported that the presence of somatic mutations in the kinase domain of EGFR correlated with increased responsiveness to EGFR-TKI.
Agree, we have added the two references.
2) Line 163: reference n°34 refers to the final OS report from the OPTIMAL trial (erlotinib vs chemotherapy), and does not reflect the improved PFS and ORR as stated by the authors. For PFS and ORR benefit of erlotinib vs chemo, the original EURTAC trial would be more appropriated (Rosell et al, The Lancet Oncol 2012; PMID: 22285168).
Agree, we have modified the text and added the reference suggested by the Reviewer.
3) Line 165: “those who”
Agree, we have corrected the sentence.
4) It is unclear how/why Figure 1 is divided into A and B… The authors should probably use color coding to discriminate between on- and off-target resistance in a single figure 1, or alternatively show a clearer separation between both mechanisms.
Agree, we have modified the figure as suggested by the Reviewer.
5) Line 260-261: The sentence “It is worth noting that loss of T790M has been observed after osimertinib treatment” is somehow confusing… By "after treatment with osimertinib" did the authors mean "after relapse to osimertinib treatment"? Loss of T790M is not surprising in response to osimertinib treatment, the interesting point is that T790M may be lost after relapse!
Agree, we have modified the text as suggested.
6) Line 395-400: The authors refer to a preprint, therefore non-peer-reviewed article, from Bivona’s lab. Although the scientific rigor and quality of this team is not questioned, it seems premature to cite these data before they have been published in a peer-reviewed journal. This reference should be removed.
7) Line 450-457: Same as before: a preprint article, although it comes from Arjun Raj's lab, should not be cited before it has been published in a peer-reviewed journal. This reference should also be removed.
We have removed the preprint from Bivona’s lab, but we’d prefer to keep the one from A. Raj, which describes a new intriguing cellular barcoding approach to investigate drug response. We believe the readers of the review will be interested by the new concept and, since we clearly indicate that it’s a preprint, they will be in a position to assess by themselves the potential significance of the study.
8) Figure 2 would need several adjustments to improve accuracy: A) arrows on the right are labeled “Resistance” whereas arrows on the left are labeled “EGFR-TKIs”, which does not make much sense… It would appear more meaningful to keep EGFR-TKIs on both sides. B) The authors state that Darwinian selection selects pre-existing tolerant cells, but seem to exclude the possibility of the selection of already resistant cells. I therefore recommend that authors replace "tolerant" with "resistant/tolerant", since current knowledge cannot exclude any hypothesis. C) For both Darwinian selection and Lamarckian induction, the authors indicate that relapsed tumors are “genetically resistant clones”. However, a significant proportion of resistance mechanisms to EGFR-TKIs is non-genetic (e.g. Epithelial-to-Mesenchymal Transition, lineage switch to Squamous, LCNEC or SCLC) and half of the resistant tumors have unknown resistance mechanism, which could therefore be either genetic or non-genetic. The authors should indicate that resistant tumors resulting from Darwinian and Lamarkian selection may also have a non-genetic resistance mechanism, or simply remove the word “genetically”.
Agree, we have modified the figure, now revised Figure 3, as suggested by the Reviewer.
9) The cellular barcoding section is probably the most interesting and original part of this manuscript compared to the many existing reviews covering the same topic, and would therefore have deserved a dedicated illustrative figure. While equivalents to figures 1 and 2 have been seen many times, a figure recapitulating the use of cellular barcoding in the context of drug-tolerance/adaptive resistance would bring a special and valuable touch of distinction to this work that would be greatly appreciated.
Agree, we have included a new Figure, revised Figure 4, to illustrate the use of cellular barcoding.
Reviewer 2 Report
Strongly agree with the thoughts expressed in the paper.
Please consider minor changes in the tet
Page 2 line 45-46
Large 45 cell carcinoma is the least frequent subtype of NSCLC and accounts for 10-15% of all
Percentage needs verification. I believe its much less.
Page 3 line 141-143
Somatic activating mutations of EGFR have
been reported in 15% of NSCLC patients, and are responsible for constitutive
ligand-independent receptor signaling [27].
I believe it is discussed below, but it is important to state here also that frequency is from 15 to above 50% depending on the population
Page 4 Table 1
I would suggest to change the title of the columns
«CNS penetration» as it does not reflect penetration, but CSF concentration due to efflux mechanisms
«EGFR mutations» according to authors idea to sensitive?
Page 5 lines 169-175
While exon
19 deletions and the L858R substitution are considered sensitizing mutations, NSCLCs
that contain EGFR exon 20 insertions are typically resistant to most EGFR-TKIs, with the
uncommon exception of the proximal A763_Y764insFQEA mutation [35]. Recently, the
novel EGFR-TKIs mobocertinib received accelerated approval from the FDA based on a
phase I/II non-randomized, open-label, multicohort clinical trial (NCT02716116) in
patients with advanced or metastatic NSCLC carrying EGFR exon 20 insertion mutations
[36,37].
I would suggest not to discuss ex20 mutations and mobivcertinib in this section
Author Response
We appreciate Reviewer 2’s positive comments, and we have revised the manuscript in accordance with his/her specific suggestions as follows:
Page 2 line 45-46, Large cell carcinoma is the least frequent subtype of NSCLC and accounts for 10-15% of all cases. Percentage needs verification. I believe its much less.
Agree, we have modified the text and added new references.
Page 3 line 141-143, Somatic activating mutations of EGFR have been reported in 15% of NSCLC patients, and are responsible for constitutive ligand-independent receptor signaling [27]. I believe it is discussed below, but it is important to state here also that frequency is from 15 to above 50% depending on the population.
Agree, we have modified the text as suggested by the Reviewer.
Page 4 Table 1, I would suggest to change the title of the columns «CNS penetration» as it does not reflect penetration, but CSF concentration due to efflux mechanisms. «EGFR mutations» according to authors idea to sensitive?
Agree, we have modified Table 1 as suggested by the Reviewer.
Page 5 lines 169-175, While exon 19 deletions and the L858R substitution are considered sensitizing mutations, NSCLCs that contain EGFR exon 20 insertions are typically resistant to most EGFR-TKIs, with the uncommon exception of the proximal A763_Y764insFQEA mutation [35]. Recently, the novel EGFR-TKIs mobocertinib received accelerated approval from the FDA based on a phase I/II non-randomized, open-label, multicohort clinical trial (NCT02716116) in patients with advanced or metastatic NSCLC carrying EGFR exon 20 insertion mutations [36,37]. I would suggest not to discuss ex20 mutations and mobivcertinib in this section.
Agree, we have modified the text as suggested by the Reviewer.
Reviewer 3 Report
The article is a wide and detailed review of Mechanisms of Acquired Resistance and Tolerance to EGFR 2 Targeted Therapy in Non-Small Cell Lung Cancer.
I have some remarks regarding this article.
The most important finding is that in my opinion the article is too long.
Despite the Editor has no limitation for article length I feel that authors too widely described all molecular mechanisms of drug resistance. Sometimes the Authors describe drug resistance not only in lung cancer but also in other tumors. This very detailed description makes the article hard to follow. I suggest simplify details and shorten the article.
Below are some minor remarks:
In introduction points 2.1 to 2.4 are not within the scope of the article and can be easily ommited to keep the article within the topic.
In table 1 Authors present FDA and China, South Korean approval. I suggest to add also EMA approval for the presented drugs.
In point 4.1in patients with exon 20 insertion not only mobocertinib but also amivantamab is approved
In point 4.2 It should be clarified thath due to irreversible action the second gen TKI`s are more efficient in treatment than first gen TKIs. Authors present clinical efficacy of 3rd gen TKIs but no information about clinical efficacy of 1st and 2nd gen TKI is presented.
In summary. The work is valuable and may be worth publication when condensed and shortened.
Author Response
We appreciate Reviewer 3’s positive comments, and we have revised and shortened the manuscript in accordance with his/her specific suggestions as follows:
In introduction points 2.1 to 2.4 are not within the scope of the article and can be easily ommited to keep the article within the topic.
Agree, we have removed the paragraphs related to surgery, radiotherapy, chemotherapy and immunotherapy.
In table 1 Authors present FDA and China, South Korean approval. I suggest to add also EMA approval for the presented drugs.
Agree, we have included EMA approval in Table 1.
In point 4.1in patients with exon 20 insertion not only mobocertinib but also amivantamab is approved
As suggested by Reviewer 2, we have shortened the paragraph related to exon 20 insertion and removed the part on mobocertinib approval.
In point 4.2 It should be clarified that due to irreversible action the second gen TKI`s are more efficient in treatment than first gen TKIs. Authors present clinical efficacy of 3rd gen TKIs but no information about clinical efficacy of 1st and 2nd gen TKI is presented.
Agree, we have included information about the relative efficacy of 1st, 2nd and 3rd generation EGFR-TKIs.
Reviewer 4 Report
1) The authors should provide a figure showing the percentages for the driver mutations in NSCLC. Additionally, the reader would benefit from highlighting the prevalence of specific mutations within the population using an infographic.
2) Cellular barcoding should be outlined in a figure format as it would help the reader visualize the entire sequence and better interpret the details provided within the review.
3) The hallmarks of drug tolerance should be reviewed. Drug-tolerant persisters (DTPs) share biological features, such as a reversible and plastic phenotype, slow-cycling capacities, and a senescent-like phenotype. The molecular mechanisms that drive the establishment and maintenance of DTPs should be discussed as it is relevant to the review.
4) The role of combinatorial therapies should be discussed. Other membrane receptors such as integrins play a vital role in the progression and recurrence of NSCLC. Please see some of the recent papers which establish the connection and the mechanism. These references need to be incorporated into the review.
a) Zhou, J., Wang, A., Cai, T. et al. Integrin α3/α6 and αV are implicated in ADAM15-activated FAK and EGFR signaling pathway individually and promote non-small-cell lung cancer progression. Cell Death Dis 13, 486 (2022). https://doi.org/10.1038/s41419-022-04928-0
b) Delahaye C, Figarol S, Pradines A, Favre G, Mazieres J, Calvayrac O. Early Steps of Resistance to Targeted Therapies in Non-Small-Cell Lung Cancer. Cancers. 2022; 14(11):2613. https://doi.org/10.3390/cancers14112613
c) Sun Q, Lu Z, Zhang Y, Xue D, Xia H, She J, Li F. Integrin β3 Promotes Resistance to EGFR-TKI in Non-Small-Cell Lung Cancer by Upregulating AXL through the YAP Pathway. Cells. 2022; 11(13):2078. https://doi.org/10.3390/cells11132078
d) Rao TC, Beggs RR, Ankenbauer KE, Hwang J, Ma VP, Salaita K, Bellis SL, Mattheyses AL. ST6Gal-I-mediated sialylation of the epidermal growth factor receptor modulates cell mechanics and enhances invasion. J Biol Chem. 2022 Apr;298(4):101726. doi: 10.1016/j.jbc.2022.101726. Epub 2022 Feb 12. PMID: 35157848; PMCID: PMC8956946.
e) Integrin as a Molecular Target for Anti-cancer Approaches in Lung Cancer.
Hassanein SS, Abdel-Mawgood AL, Ibrahim SA. EGFR-Dependent Extracellular Matrix Protein Interactions Might Light a Candle in Cell Behavior of Non-Small Cell Lung Cancer. Front Oncol. 2021 Dec 15;11:766659. doi: 10.3389/fonc.2021.766659. PMID: 34976811; PMCID: PMC8714827.
Author Response
We appreciate Reviewer 3’s positive comments, and we have revised and shortened the manuscript in accordance with his/her specific suggestions as follows:
1) The authors should provide a figure showing the percentages for the driver mutations in NSCLC. Additionally, the reader would benefit from highlighting the prevalence of specific mutations within the population using an infographic.
Agree, we have included a new figure, revised Figure 1, to illustrate the percentages of the driver oncogenes in NSCLC.
2) Cellular barcoding should be outlined in a figure format as it would help the reader visualize the entire sequence and better interpret the details provided within the review.
Agree, we have included a new figure, revised Figure 4, to illustrate cellular barcoding.
3) The hallmarks of drug tolerance should be reviewed. Drug-tolerant persisters (DTPs) share biological features, such as a reversible and plastic phenotype, slow-cycling capacities, and a senescent-like phenotype. The molecular mechanisms that drive the establishment and maintenance of DTPs should be discussed as it is relevant to the review.
The hallmarks of DTPs and the underlying molecular mechanisms are extensively discussed in the longest paragraph of our manuscript (5. Drug tolerant persister states). Since other Reviewers found the article too long, we believe it wouldn’t be suitable to further develop this section.
4) The role of combinatorial therapies should be discussed. Other membrane receptors such as integrins play a vital role in the progression and recurrence of NSCLC. Please see some of the recent papers which establish the connection and the mechanism. These references need to be incorporated into the review.
We discussed the use of combinatorial therapies for certain mechanisms of resistance/tolerance, such as AURKA, YAP and ferroptosis. As recommended by the Reviewer, we included a new reference about the role of integrins. However, considering the extremely vast literature on the different mechanisms potentially involved in NSCLC resistance to EGFR-TKIs, we believe that adding further details on integrins would be beyond the scope of our review.